# SerpinA3K Deficiency Reduces Oxidative Stress in Acute Kidney Injury

**DOI:** 10.3390/ijms24097815

**Published:** 2023-04-25

**Authors:** Isaac González-Soria, Axel D. Soto-Valadez, Miguel Angel Martínez-Rojas, Juan Antonio Ortega-Trejo, Rosalba Pérez-Villalva, Gerardo Gamba, Andrea Sánchez-Navarro, Norma A. Bobadilla

**Affiliations:** 1Molecular Physiology Unit, Instituto de Investigaciones Biomédicas, Universidad Nacional Autónoma de México, Mexico City 04510, Mexico; 2Department of Nephrology and Mineral Metabolism, Instituto Nacional de Ciencias Médicas y Nutrición Salvador Zubirán, Mexico City 14080, Mexico; 3PECEM (MD/PhD), Facultad de Medicina, Universidad Nacional Autónoma de México, Mexico City 04510, Mexico

**Keywords:** renal ischemia/reperfusion, FOXO3, HIF1a

## Abstract

We previously showed that SerpinA3K is present in urine from rats and humans with acute kidney injury (AKI) and chronic kidney disease (CKD). However, the specific role of SerpinA3K during renal pathophysiology is unknown. To begin to understand the role of SerpinA3K on AKI, SerpinA3K-deficient (KOSA3) mice were studied 24 h after inducing ischemia/reperfusion (I/R) and compared to wild type (WT) mice. Four groups were studied: WT+S, WT+IR, KOSA3+S, and KOSA3+IR. As expected, I/R increased serum creatinine and BUN, with a GFR reduction in both genotypes; however, renal dysfunction was ameliorated in the KOSA3+IR group. Interestingly, the increase in UH_2_O_2_ induced by I/R was not equally seen in the KOSA3+IR group, an effect that was associated with the preservation of antioxidant enzymes’ mRNA levels. Additionally, FOXO3 expression was initially greater in the KOSA3 than in the WT group. Moreover, the increase in BAX protein level and the decrease in *Hif1a* and *Vegfa* induced by I/R were not observed in the KOSA3+IR group, suggesting that these animals have better cellular responses to hypoxic injury. Our findings suggest that SerpinA3K is involved in the renal oxidant response, HIF1α/VEGF pathway, and cell apoptosis.

## 1. Introduction

Acute kidney injury (AKI) is a worldwide concern since around half of the patients in the intensive care unit (ICU) develop this disease during their hospital stay, which is also associated with a higher mortality rate [1]. AKI is characterized by an abrupt reduction in kidney function in less than 7 days, which is evidenced by increased serum creatinine levels or reduced urinary output [2]. The impact of AKI on mortality was recently shown in the SARS-CoV-2 pandemic, in which patients with COVID-19 and AKI had mortality rates between 23% and 50% [3]. Additionally, it is well recognized now that AKI survivors may develop chronic kidney disease (CKD) in the subsequent months [4]. The most common cause of AKI is renal blood flow reduction [5] with resulting renal ischemia that causes an imbalance in oxygen supply and energetic demand [6].

During AKI, the proximal tubular epithelium is exposed to hypoxic conditions [7], leading to injury and cell death that are worsened after reperfusion due to reactive oxygen species (ROS) generation [6]. Coupled with this, there is a recruitment of neutrophils, monocytes/macrophages, dendritic cells, and natural killer lymphocytes through damage-associated molecular patterns (DAMPs) and chemokines [6,7,8]. In addition, mitochondrial disturbances contribute to ROS generation and promote cell death [7,9]. Despite advances in understanding the pathophysiology of AKI, many proteins and signaling pathways remain to be explored.

We previously reported that SerpinA3K is an early biomarker of AKI and AKI to CKD transition in rats due to an upregulation in SerpinA3K expression. In the case of AKI to CKD transition, the abnormal presence of SerpinA3K in urine preceded the increase in serum creatinine, urinary protein excretion, and kidney fibrosis. Furthermore, the abnormal presence of SerpinA3 was found in urine samples from patients with lupus nephritis, focal and segmental glomerulosclerosis, and ANCA-associated vasculitis. We also showed that SerpinA3K is expressed in the renal tubular epithelium and that in AKI and AKI to CKD transition in the rat, as well as in patients with lupus nephritis, it is relocated from the cytosol to the tubular apical membrane, suggesting that it is secreted during renal damage. However, the role of SerpinA3K remains elusive [10,11,12].

Serpins are a family of serine protease inhibitors (SERPINs); therefore, these proteins are found in the intracellular and extracellular compartments. The serpin family is composed of 34 members that possess a central domain with three folded beta sheets and 8–9 alpha helices; their capacity as inhibitors of serine proteases depends on this domain. The molecular weight is around 46–55 kD. SerpinA3 has been identified as a specific inhibitor of tissue kallikrein and named kallikrein-binding protein, which is expressed in the liver, pancreas, kidney, cornea, and skin. Human SerpinA3 is homologous to SerpinA3K in rodents [12].

Zhang B. et al. and Liu X et al. showed that SerpinA3K exerts other non-canonical functions, such as: blocking the generation of reactive oxygen species (ROS), reducing inflammation, and inhibiting the activation of pro-fibrotic pathways in the damage induced to the cornea and retina [13,14,15,16,17,18,19,20]. In addition to this, we recently demonstrated in HEK-293 cells that SerpinA3K is secreted under noxious stimuli, such as nutrient deprivation and exposure to hydrogen peroxide [21].

The overexpression and urinary excretion of SerpinA3K in animals and patients with AKI and CKD suggest that this protein may have an active role during kidney damage. However, the consequence of SerpinA3K deficiency in basal and in AKI conditions has not been reported. Therefore, this study was designed to address the biological function of SerpinA3K in the kidney, using SerpinA3K knockout mice to evaluate the effect of its absence on renal function regulation and during the development of ischemia/reperfusion (I/R)-induced AKI.

## 2. Results

### 2.1. Characterization of SerpinA3K Deficiency in Mice

To the best of our understanding, we are the first group to describe the effect of SerpinA3K deficiency in basal kidney function and during AKI conditions. Figure 1A depicts the genotyping for wild type (WT), heterozygous (HT), and knockout SerpinA3K (KOSA3) mice, in which the deletion of SerpinA3K is clearly demonstrated in the KOSA3 mice. SerpinA3K expression was evaluated by Western blot in the brain, kidney, liver, heart, and lung in WT and KOSA3 mice (Figure 1B). In the WT mice, SerpinA3K is expressed in all studied tissues, the brain being the one that shows less expression. As expected, this tissue expression pattern was not seen in the KOSA3 mice. Once, SerpinA3K deficiency in the KOSA3 mice was corroborated, we evaluated the expression of this protein in the kidneys from the WT, HT, and KOSA3 mice. As shown in Figure 1C, WT and HT mice exhibited similar SerpinA3K expression. Thus, for the purpose of this study, only the WT and the KOSA3 mice were included and studied in the following experiments. Moreover, SerpinA3K deficiency did not impact the mortality of these mice during the follow-up for 18 months (Figure 1D).

### 2.2. Impact of SerpinA3K Deficiency during AKI

Renal injury was evaluated 24 h after bilateral renal ischemia, as is shown in Figure 2 and Figure 3. Left and right kidney weights were similarly increased in the WT+IR and in the KOSA3+IR groups (Figure 2A,B). The ischemic injury was associated with a significant increase in FENa, an effect that was similar in both genotypes (Figure 2C). In contrast, the magnitude in the elevation of serum creatinine and BUN in the WT+IR group was higher than in the KOSA3+IR group (Figure 2D,E). These findings were confirmed analyzing the GFR by FITC-sinestrin clearance. The percentage change in GFR appears in Figure 2F, where it is evident that the reduction in GFR was 75% in the WT+IR group, while in the KOSA3+IR group, it was 55% (*p* = 0.0034). These results show that the kidney dysfunction induced by I/R was milder in the KOSA3+IR compared to the WT+IR group.

### 2.3. Histopathological Evaluation

After functional measurements, we evaluated morphological changes at 24 h of reperfusion in all studied groups, using two independent scores corresponding to tubular injury and tubular necrosis. Figure 3A,B show representative microphotographs of the cortex for the WT+IR and KOSA3+IR groups, respectively, while Figure 3C,D show the renal medulla for the WT+IR and KOSA3+IR groups, respectively. Similar and significant tubular abnormalities were observed in both cortex and medulla regions, including loss of brush border, tubular dilation, cytoplasmatic vacuolar degeneration, and even hyaline casts in the WT+IR and KOSA3+IR groups (Figure 3A–F). As expected, we observed an apparent increase in cellular necrosis both in the cortex and medulla of the WT+IR groups. Nevertheless, there was a considerably higher frequency of this response in the medullar fields (Figure 3A–D,G,H). Surprisingly, we did not find an independent effect of the genotype on these morphological items, suggesting a minor role of SerpinA3K on cellular epithelial injury and cell death in this model.

### 2.4. Renal Location of SerpinA3K and Renal Damage Evaluation of Acute Kidney Injury

In addition, to the histopathological findings, urinary biomarkers of AKI were assessed. As we previously reported, urinary SerpinA3K excretion (uSA3K) was detected in the WT+IR group but not in the WT+S group. The expected absence of uSA3K in the KOSA3 groups was confirmed (Figure 4A). In Figure 4B are representative microphotographs of SerpinA3K immunofluorescence for the WT+S, WT+IR, and KOSA3+IR groups. In the WT+S mice, SerpinA3K was mainly observed in the cytosol of tubular epithelial cells. This expression was more evident in the WT+IR group, together with the relocation to the apical membrane, as we previously reported in rats and humans with AKI and CKD diseases [10,11,21]. As expected, no immunostaining was observed in the KOSA3+IR group. According to our histopathological findings, urinary biomarkers of AKI: KIM1 and HSP72 increased significantly in both the WT+IR and the KOSA3+IR groups (Figure 4C,D). Finally, cellular apoptosis was indirectly evaluated by assessing the pro-apoptotic protein BAX. As expected, BAX expression was significantly increased in the WT+IR group, but such an increment was not seen in the KOSA3+IR group (Figure 4E).

### 2.5. Inflammatory Response during AKI in SerpinA3K Deficiency

The immune response was evaluated through pro-inflammatory cytokines mRNA levels like *Il6, Tnf-alpha,* and *Ccl2.* We only observed a significant elevation of *Il-6* in both the WT+IR and the KOSA3+IR groups (Figure 5A–C). At the same time, reparative cytokines were evaluated. Ischemic injury induced an increase in *Tgfb1* in the WT+IR and KOSA3+IR groups with no difference in *Il-10* (Figure 5D,E). It is well known that inflammation can be regulated by the Wnt-β-catenin pathway [22]. Under normal conditions, it is continuously degraded, but it is preserved during cellular stress. Therefore, β-catenin expression was explored; however, no differences were observed among the studied groups, as is shown in Figure 5H. Additionally, it has been reported that in the initial response to I/R there is an association between β-catenin and HIF-1α, the major responder to hypoxia. *Hif1a* was only upregulated in the KOSA3+IR group (Figure 5F) [23]. In correspondence to this finding, *Vegfa,* that is a target of *Hif1a,* was upregulated in the KOSA3+IR, while the WT+IR group showed a down-regulation of *Hif1a* and *Vegfa* (Figure 5F,G).

### 2.6. Response to Oxidative Stress in SerpinA3K Deficient Mice

As mentioned above, hypoxia and consequently ROS generation play a key role in the pathogenesis of AKI. Urinary hydrogen peroxide excretion (UH_2_O_2_) usually increases under oxidative stress conditions [24]. We found that UH_2_O_2_ was significantly higher in the WT+IR mice. However, basal UH_2_O_2_ was higher in the KOSA3+S than the WT+S, but it was not increased even more in the KOSA3+IR mice, suggesting that there is a difference in redox homeostasis in these mice even in basal conditions (Figure 6A). The Nuclear Factor Erythroid-derived 2-like 2 (*Nfe2l2*, also known as *Nrf2*) is a central orchestrator of the redox balance. In stress conditions, it is translocated into the nucleus to induce the expression of antioxidant enzymes, including catalase and superoxide dismutase (SOD) [25]. No differences in *Nfe2l2* were found, but a significant reduction in *Cat*, *Sod2*, and glutathione peroxidase 1 (*Gpx1*) mRNA levels were observed in the WT+IR group [26]. Intriguingly, the downregulation of these transcripts was not observed in the KOSA3+IR group (Figure 6B–E).

Moreover, it has been reported that Sirtuin-1 protects against ROS [27]. In this regard, we found that the WT+IR group had a significant increase in Sirtuin-1 expression, suggesting that this response is due to the high amount of ROS generated during this injury. Interestingly, such an increment was not seen in the KOSA3+IR group (Figure 6F). Furthermore, the transcription factor Forkhead 3 (FOXO3) and the peroxisome proliferator-activated receptor gamma co-activator 1 alpha (PGC1-α) were evaluated as targets of ischemic injury and Sirtuin-1 (Figure 6G,H). In basal conditions, FOXO3 was significantly elevated in the KOSA3+S group compared to the WT+S group, but FOXO3 expression was not altered when both genotypes underwent I/R. In spite of Sirtuin-1 elevation in the WT+IR group, PGC1-α was downregulated compared to the WT+S group. No differences between the KOSA3 and KOSA3+IR groups were found.

## 3. Discussion

Recently, SerpinA3 has attracted the attention of various research groups due to its multiple canonical and non-canonical roles in several diseases; however, there is not enough evidence on the physiological and pathophysiological role of SerpinA3K in renal tissue [12]. To the best of our knowledge, this is the first analysis of renal injury induced by I/R in SerpinA3K deficient mice. First, we validated that the SerpinA3K knockout mice exhibit a complete deficiency of this protein in different tissues such as the brain, liver, heart, lung, and kidney. Second, the deletion of this protein did not have any impact on the survival of mice of at least 18 months of age; however, long-term organ function in KOSA3 mice remains to be elucidated. Third, after the ischemic injury, an attenuation in the renal dysfunction induced by I/R was observed, together with a remarkable regulation of genes related to the redox system when SerpinA3K was absent.

In the context of bilateral I/R-induced AKI, an attenuation in GFR reduction in the KOSA3+IR mice was observed when compared to the WT+IR group (53.9 vs. 76.4%, respectively, *p* = 0.0034), suggesting that SerpinA3K is involved in the renal hemodynamic response to I/R. Studies performed in animals with AKI indicate that the sustained reduction in renal blood flow and GFR is a consequence of overactivation of the tubuloglomerular feedback, secondary to proximal tubular necrosis [28]. It would be interesting in future studies to evaluate the tubuloglomerular feedback, the SNGFR, and the morphology of the proximal tubular epithelium in these groups of animals, especially since SNGFR and the morphology can now be evaluated by linescan multiphoton microscopy [29]. Interestingly, we confirmed that the expression of SerpinA3K increased after AKI, specifically in tubular cells, where it was relocated to the apical membrane [21]. The main structure affected by AKI is the tubular epithelium, so our results suggest that SerpinA3K deficiency attenuated the drop in GFR and may result from a more preserved tubular epithelium. Although we did not find changes in global necrosis, the difference between BAX expression after I/R suggests greater death by apoptosis in the WT+IR mice than in the KOSA3+IR group. Nevertheless, we cannot exclude other types of cell death.

To provide further evidence for this fact, we evaluated the histological changes in tubular architecture and the excretion of biomarkers of AKI. Surprisingly, we did not observe a notorious reduction in any of these parameters. The inflammatory response activation for the I/R injury was evidenced by the upregulation of *Il6* and *Tgfb1* mRNA levels in the WT+IR and KOSA3+IR groups [30], with no difference between genotypes. Nevertheless, SerpinA3K has been associated with an anti-inflammatory effect in the retina and cornea, in which one of the possible mechanisms is through the canonical Wnt pathway [14,15,17,19]. Briefly, the Wnt pathway during the transient activation of β-catenin can upregulate various genes in different diseases to trigger cellular repairment, but its persistence might generate injury [22]. It has been reported that SerpinA3K blocks the co-receptor LRP6 necessary to activate Wnt signaling [31]. However, in the acute stage of our study, the involvement of β-catenin was not seen. According to Xu ZH. et al. [23], the increase in β-catenin is reached 7 days after I/R. Therefore, we cannot exclude the role of this protein in the subsequent days. Additionally, we observed a significant increase in *Hif-1a* levels in the KOSA3+IR group, which has been correlated with the activity of Wnt-β-catenin to promote the reparative process after AKI [23]. Consistent with the fact that *Vegfa*, a gene target of β-catenin and HIF-1α, is downregulated when SerpinA3K is overexpressed, in this study, we observed the opposite effect in the KOSA3+IR group [14,15,17,19]. Furthermore, it is known that the early activation of HIF-1α and the overexpression of VEGFA may promote repair after an AKI episode [23,32]. These findings suggest that SerpinA3K regulation of *Vegfa* during AKI is not only through the Wnt-β-catenin pathway.

According to the previous findings, the response to hypoxia could be different in the absence of SerpinA3K. We observed a remarkable change in the UH_2_O_2_ in the KOSA3 mice, suggesting a relevant role of SerpinA3K in redox renal pathophysiology [24]. Unexpectedly, we found that the absence of SerpinA3K was associated with an improvement in *Cat*, *Sod2*, and *Gpx1* expression after I/R, suggesting a limited antioxidant response during AKI when SerpinA3K is present. These findings were surprising considering that SerpinA3K has been reported to protect against oxidative stress in the cornea and retina [13,14,20]. Nevertheless, the levels of UH_2_O_2_ in basal conditions could lead to a better adaptation to increased levels of ROS in the KOSA3 mice, which become more evident after hypoxia and reoxygenation are induced. This could also be related to a higher expression of FOXO3, as observed, which could be a condition to optimize the antioxidant response under physiological conditions.

It has been reported that the deacetylase Sirtuin-1 is a stress-responsive protein that protects against ROS in mice with AKI, attenuating the damage of renal cells. [27,33]. In this sense, it has been reported that Sirtuin 1 overexpression is detected 24 h after I/R, but its expression is enhanced even more after 7 days, suggesting its participation in kidney repair [34]. We confirm this finding in the WT mice, but it was not found in the KOSA3+IR mice, suggesting that renal injury generated by I/R was milder in the absence of SerpinA3K.

The transcription factor FOXO3 plays an important role in the resistance against oxidative stress induced by AKI, through inducing antioxidant enzymes [35,36]. Interestingly, at baseline, we observed that the expression of FOXO3 in the KOSA3+S group was higher than in the WT+S group. These results strongly suggest that SerpinA3K-deficient mice are initially better prepared to respond against a deleterious insult, while PGC1-α, a main regulator of mitochondrial biogenesis, is reduced after an I/R episode [37]. In this study, a significant reduction in PGC-1α in the WT+IR group was confirmed. However, in the KOSA3+IR group, the PGC-1α expression was not reduced compared to the KOSA3 group, again suggesting that the cellular homeostasis is different during AKI in these mice.

The discrepancy in the regulation of Sirtuin 1 and FOXO3 on PGC-1α and BAX could be explained by the timing of when these proteins were studied, which was only 24 h after the I/R. Therefore, more studies are necessary to evaluate the temporal course of these molecules, as well as their interrelation, on the long-term consequences of AKI.

One limitation of this study is that it was not designed to assess the long-term consequences of AKI but only the short-term ones, such as 24 h after I/R. The other limitation is that we cannot exclude that other mouse SerpinA3 isoforms may compensate for the absence of SerpinA3K.

Our results suggest that the higher levels of antioxidant mRNA, and higher expression of FOXO3, without the increased expression of BAX, despite the lack of regulation by Sirtuin 1, reflect the optimized antioxidant response in the KOSA3+IR group.

## 4. Material and Methods

All experiments involving animals were conducted in strict accordance with the NIH Guide for the Care and Use of Laboratory Animals and with the Mexican Federal Regulation for animal reproduction, care, and experimentation (NOM-062-ZOO-2001). The study was approved by the Animal Care and Use Committee of the Instituto Nacional de Ciencias Médicas y Nutrición Salvador Zubirán (NMM-1984-19-22-1), Mexico City. SerpinaA3K(−/−) knockout (KOSA3) mice were acquired in The Jackson Laboratory under a genetic background C57BL/6NJ. In brief, these KOSA3 mice were generated by the deletion of 580 bp into exon 4 of the serine peptidase inhibitor, clade A, member 3K (SerpinA3K gene). For more details: https://www.jax.org/strain/031522 (accessed on 21 March 2023). All the animals were housed in ventilated racks, 2–3 per cage (NexGen™ Mouse 500. Ecoflo, Allentown, PA, USA) and on a 12:12 h light–dark cycle at temperature of 20–21 °C and humidity of 29–33%. All mice had free access to water and standard food.

### 4.1. Mouse Kidney Bilateral Renal Ischemia Model

Forty male mice aged 3 months were included, of which twenty were wild type (WT, (SerpinA3K(+/+*)*)) and twenty knockout for SerpinA3K(−/−) (KOSA3). All mice included were assigned using the random number calculator by GraphPad. One half of each genotype were subjected to sham surgery (WT+S and KOSA3+S, respectively) and the other half of each genotype underwent renal bilateral ischemia for 30 min and 24 h of reperfusion for (WT+IR and KOSA3+IR, respectively). For this purpose, the mice were anesthetized using an i.p. injection of sodium pentobarbital (50 mg/kg). After they were sedated, the back was shaved and located in a heating pad (SurgiSuite; Kent Scientific Corporation, Torrington, CT, USA) to control temperature at 37 ± 0.4 °C. Bilateral flank incisions were made to expose the kidneys outside the cavity and the renal pedicle was dissected. Then, non-traumatic vascular clamps were collocated over the pedicles for exactly 30 min and then released for reperfusion, which was confirmed by the color recovery of the kidney. The mice were rehydrated with 0.5 mL 0.9% NaCl. The incision was closed in 2 layers with 4-0 silk sutures. In the case of sham surgery, the procedure was the same but without clamping pedicles. After waking, they were located in metabolic cages for 18 h urine collection (from 15:00–9:00 h). The next day, glomerular filtration rate was evaluated, and then the mice were euthanized after being sedated and cardiac puncture was performed to take blood samples and their kidneys were isolated. One kidney was stored at −70 °C for molecular and biochemical analyses and the other kidney was fixed with formaldehyde 4% and then kept in holding buffer until the paraffin inclusion.

### 4.2. Glomerular Filtration Rate Measurement

Glomerular filtration rate (GRF) was determined as described before [38,39]. Briefly, mice were anesthetized with sodium pentobarbital (30 mg/kg). This dose was chosen to keep mice sedated but without impact on GFR. Afterwards, a transdermal monitor was located and fixed on the back. First, a basal reading was made during 2 min and then a FITC-sinestrin (35 mg/mL dissolved in 0.9% NaCl) was injected into the retroorbital sinus (7 mg/kg BW) and FITC-sinestrin clearance rate was measured for 1 h.

### 4.3. Histological Assessment of Tubular Injury

Kidney slices of 4 µm were stained with periodic acid-Schiff (PAS) to evaluate tubular damage. Score damage was evaluated according to McLarnon, SR et al. with slight modifications (Appendix A) [40]. Briefly, six high-power fields in the cortex and in the corticomedullary junction (magnification 200×) were recorded and scored. Tubular injury was evaluated taking into account loss of brush border, intratubular casts, tubular dilation, and vacuolar degeneration. Each field was scored separately with 0 = non-injury, 1 = 1–25% of area affected, 2 = 26–50%; 3: 50–75%; 4: 76–100%. Necrosis was evaluated separately with the same score described before.

### 4.4. SerpinA3K Immunofluorescence

We performed the same procedure as described before [41]. Briefly, mice kidney tissue embedded in paraffin was used to obtain slices of 4 µm that were then deparaffined. The process for antigen recovery was made with citrate buffer (Cat. No. BSB 0022, Bio SB) and then the slices were blocked with 10% bovine serum albumin (BSA). Primary antibody anti-SerpinA3K (1:500; Proteintech, Cat. No. 554480-1-AO) was used and anti-rabbit Alexa Fluor-488 (1:1000, Cat. No. A32723, ThermoFisher, Waltham, MA, USA) was used as secondary antibody. Zeiss LSM 710 Duo (Oberkochen, Germany) was used to take the photos 40×.

### 4.5. Urinary Hydrogen Peroxide Excretion

Urinary hydrogen peroxide concentration was measured by using a commercial kit (Amplex Red Hydrogen Peroxide/Peroxidase Assay, cat. No. A22188, ThermoFisher Scientific, Waltham, MA, USA) following manufacturer’s instructions.

### 4.6. Serum and Urine Creatinine

Creatinine was measured with a commercial kit (LabAssay TM Creatinine (Jaffé method) FUJIFILM Wako Shibayagi Corporation) according to the manufacturer’s instructions.

### 4.7. Serum BUN, Sodium, and Potassium

Serum BUN and electrolyte concentration were automatically evaluated with a SYNCHRON System (Beckman Coulter, Brea, CA, USA).

### 4.8. Western Blot

Urinary biomarkers were evaluated as we described before [38] with little modifications, in the case of urine from mice, 3 µL were used. Briefly, the proteins were blotted in a PVDF membrane, and each membrane was blocked with a 5% blocking agent for 90 min, then they were incubated overnight at 4 °C with anti-KIM-1 (1:1000; Boster, Cat. No. PA1632, Pleasanton, CA, USA) or with anti-SerpinA3K (1:2000; Proteintech, Cat. No. 554480-1-AO). Next, they were incubated with secondary anti-rabbit (1:2500; Sigma-Aldrich, Cat. No. A0545). Finally, the bands were identified by using a chemiluminescence kit (Millipore, Cat. No. WBKLS0500, Burlington, MA, USA) and detected in iBrightFL1500 (ThermoFisher Scientific, Waltham, MA, USA).

Portions of brain, kidney, liver, lung, and heart were homogenized with a lysis buffer (50 mM HEPES pH 7.4, 250 mM NaCl, 5 mM EDTA, 0.1% NP-40) and complete protease inhibitor (Roche, Cat. No. 11697498001). Protein concentration was measured by Lowry protein assay (Bio-Rad, Cat. No. 5000113 and 5000114). Proteins were denatured and electrophoresed in 8.5% or 12% acrylamide gel, blotted into a PVDF membrane, and incubated with anti-SerpinA3K(1:2500; Proteintech, Cat. No. 554480-1-AO), anti-SIRT1 (1:1500; Santa Cruz, Cat. No. sc-74504), anti-FOXO3 (1:2000; Santa Cruz, Cat. No. sc-11351), PGC-1α (1:1000; Invitrogen, Cat. No. PA5-38021), BAX (1:2000; Sigma-Aldrich, Cat. No. SAB5701333), β-catenin (1:1000; Cell signaling, Cat. No. 8480S), and HRP β-actin (1:1,000,000; Abcam, Cat. No. ab49900) overnight at 4 °C. Then, they were incubated with their respective secondary antibody: anti-rabbit (Sigma, A0545) or anti-mouse (Jackson, 115-035-174). The bands were detected as described above. The optical density was measured with ImageJ.

### 4.9. mRNA Levels Assessed by Semiquantitative RT-PCR

Total RNA was isolated using a TRIzol reagent (ambiol) and checked for integrity in an agarose gel electrophoresis and by spectrometry. Reverse transcription was performed to obtain cDNA. The probes used are listed in Table 1. The mRNA levels were quantified by real-time PCR on a QuantStudio 5 (ThermoFisher Scientific, Waltham, MA, USA). The relative quantification of each gene expression was performed with the comparative threshold cycle method (Ct).

### 4.10. Statistical Analysis

GraphPad prism version 8.0.1 was used to perform statistical analysis and graphs. The data distributions are represented as mean ± IC95%. The data that accomplish ANOVA assumptions were compared with one-way ANOVA and the rest were compared with Kruskal–Wallis. Post hoc analysis was performed with Tukey’s multiple comparisons test. Statistical significance was defined as *p*-value < 0.05.

## 5. Conclusions

This study is the first to evaluate the impact of SerpinA3K deficiency in the setting of AKI. In spite of severe renal injury induced by 30 min of I/R, the initial molecular response in the absence of SerpinA3K was different than in WT mice. Our results show that the attenuation in renal dysfunction after I/R observed in the KOSA3+IR group seems to be mediated by a better response against renal hypoxia. This appears to be mediated by an early upregulation of HIF-1α and sustained regulation of antioxidant enzymes, which could be promoted by a higher initial expression of intrarenal FOXO3. The antioxidant protection observed was associated with diminished pro-apoptotic signaling. Further studies are required to address the relevance of SerpinA3K deficiency in different pathological settings, including CKD, diabetic nephropathy, and even immune-mediated nephritis.

## Figures and Tables

**Figure 1 ijms-24-07815-f001:**
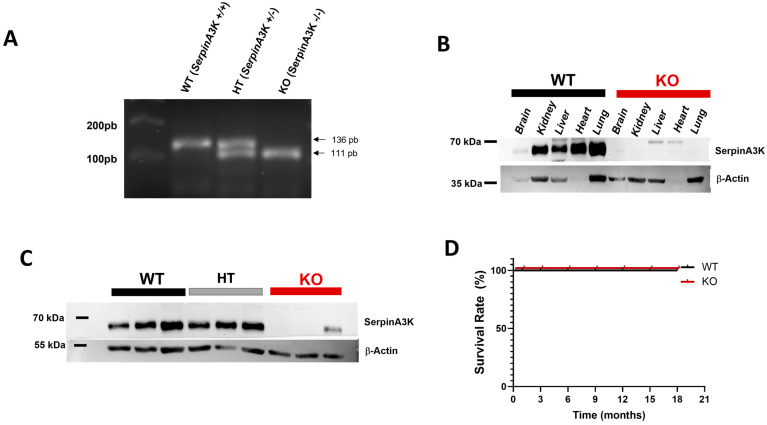
**Characterization of SerpinA3K-deficient mice.** (**A**) Genotyping for WT, HT, and KOSA3 mice, (**B**) SerpinA3K expression in brain, kidney, liver, heart, and lung, (**C**) kidney expression of SerpinA3K. (**D**) Survival rate in the following 18 months.

**Figure 2 ijms-24-07815-f002:**
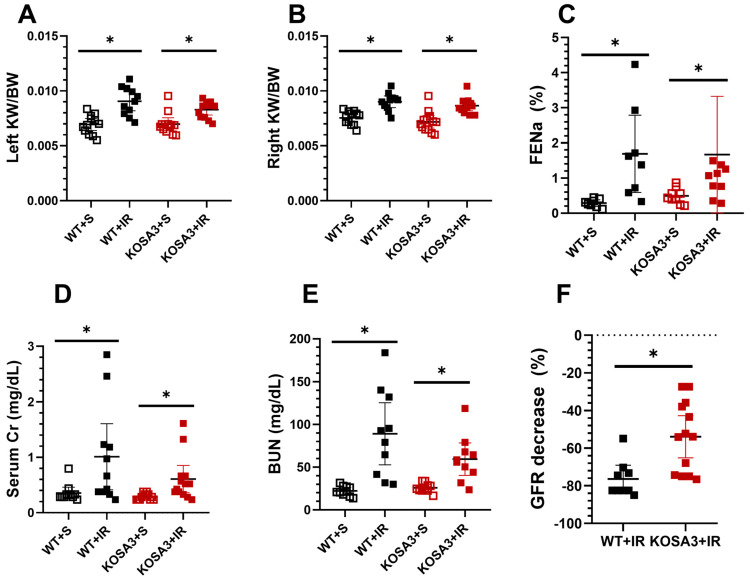
**Impact of SerpinA3K deficiency during AKI.** (**A**,**B**) Left and right kidney normalized by body weight, (**C**) fractional excretion of sodium (FENa), (**D**) serum creatinine, (**E**) blood urea nitrogen (BUN), (**F**) percentage decrease in glomerular filtration rate (GFR) in the WT+IR and KOSA+IR groups compared with their respective control, the WT+S and KOSA3+S groups. *n* = 8–13 per group. * *p* < 0.05 vs. the group stated by the upper solid bar. WT+S: WT [SerpinA3K(+/+)] + sham surgery, WT+IR: WT + ischemia/reperfusion of 30 min, KOSA3+S: KO [SerpinA3K(−/−)] + sham surgery, KOSA3+IR: KO + ischemia/reperfusion of 30 min. KW: kidney weight, BW: body weight.

**Figure 3 ijms-24-07815-f003:**
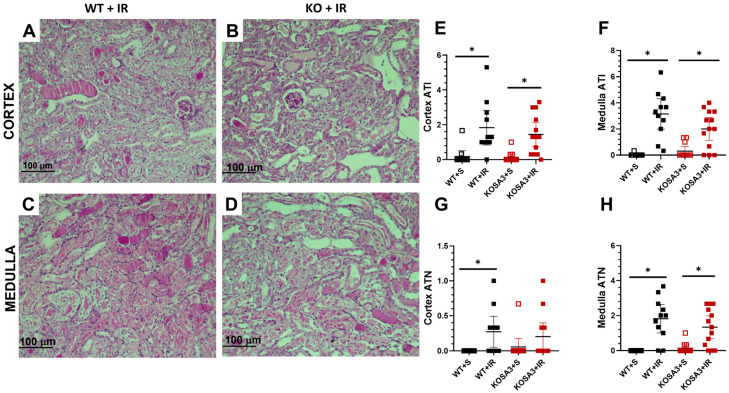
**Histopathological evaluation.** (**A**–**D**) Representative microphotographs of kidney cortex (upper panel) and medulla (lower panel) for the WT+IR and KOSA3+IR groups, respectively, 24 h after I/R. (**E**,**F**) Acute tubular injury (ATI) score for kidney cortex and medulla. (**G**,**H**) Acute tubular necrosis (ATN) score for kidney cortex and medulla. *n* = 10–13 per group. * *p* < 0.05 vs. the group stated by the upper solid bar.

**Figure 4 ijms-24-07815-f004:**
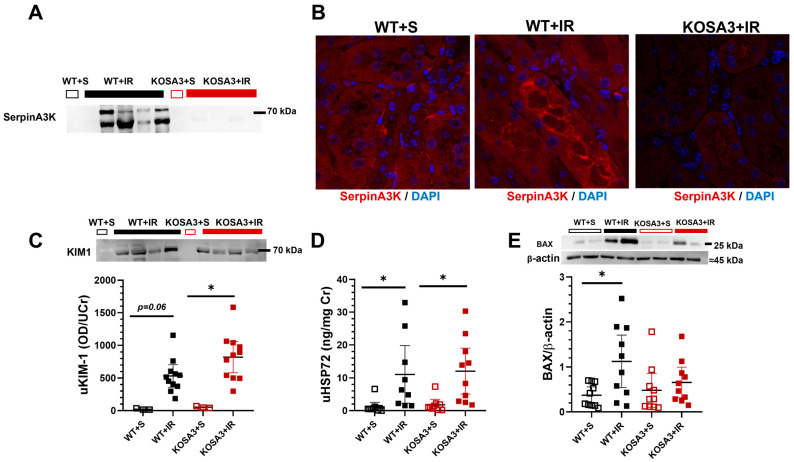
**Evaluation of renal damage 24 h after inducing ischemia/reperfusion.** (**A**) A representative Western blot for urinary SerpinA3K excretion, (**B**) immunofluorescence staining for SerpinA3K, (**C**) urinary KIM-1/urinary creatinine ratio, (**D**) urinary HSP72/urinary creatinine ratio, (**E**) and representative Western blot of BAX. *n* = 8–11 per group. * *p* < 0.05 vs. the group stated by the upper solid bar.

**Figure 5 ijms-24-07815-f005:**
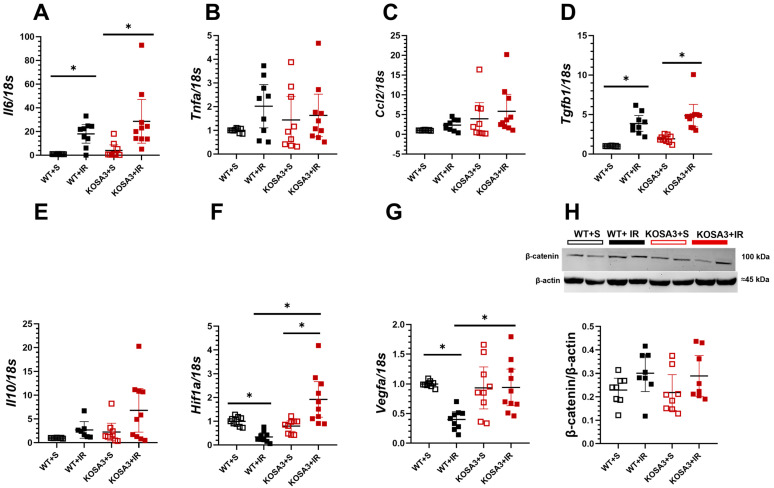
**Inflammatory response during AKI in SerpinA3K deficiency.** mRNA levels of (**A**) *Il6*, (**B**) *Tnfa*, (**C**) *Ccl2*, (**D**) *Tgfb*, (**E**) *Il10***,** (**F**) *Hif1a*, (**G**) *Vegfa*, (**H**) a representative Western blot image expression of β-catenin. *n* = 8−11 per group. * *p* < 0.05 vs. the group stated by the upper solid bar.

**Figure 6 ijms-24-07815-f006:**
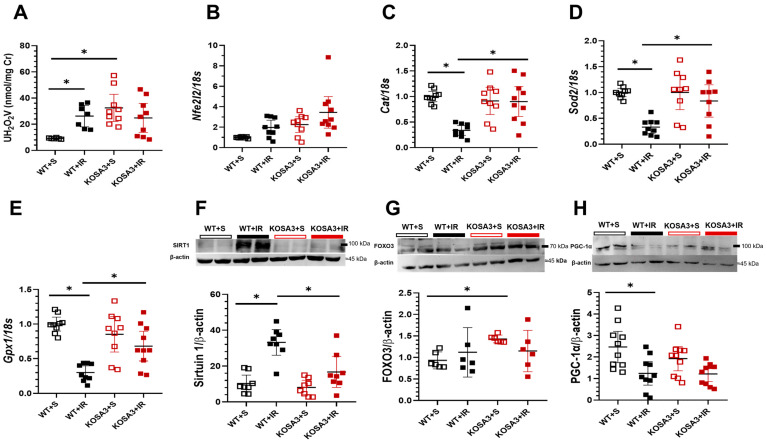
**Response to oxidative stress.** (**A**) Urinary hydrogen peroxide excretion/creatinine ratio, UH_2_O_2_ (**B**) *Nfel2l2*, (**C**) *Cat*, (**D**) *Sod2*, (**E**) *Gpx1* mRNA levels. Representative Western blots for (**F**) Sirtuin 1, (**G**) FOXO3**,** and (**H**) PGC-1α. *n* = 6–11 per group. * *p* < 0.05 vs. the group stated by the upper solid bar.

**Table 1 ijms-24-07815-t001:** Taqman probes used for the amplification of each specific gene.

Target Genes	Probe gen Catalog Number
*Hif1a*	Mm00468869_m1
*Gpx1*	Mm00656767_g1
*Sod2*	Mm01313000_m1
*Tnfa*	Mm00443258_m1
*Il6*	Mm00446190_m1
*Tfgb1*	Mm01178820_m1
*Il10*	Mm01288386_m1
*Ccl2*	Rn00580555_m1
*Nfe2l2*	Rn00582415_m1
*Vegfa*	Rn01511602_m1

## Data Availability

The data that support the findings of this study are available on request from the corresponding author (NAB).

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
