# Peer review of "SerpinA3K Deficiency Reduces Oxidative Stress in Acute Kidney Injury"

_ijms, 2023, doi:10.3390/ijms24097815_

Round 1

Reviewer 1 Report

In this study the authors evaluated the renal function in KOSA3 mice (deficient of SerpinA3K) to establish the role of SerpinA3K during Acute kidney injury. First, renal histology and biochemical assays were carried out during 24h reperfusion post AKI to comparerenal function in KOSA3 compared to WT mice. The results showed an improvement in renal function in KOSA3 mice, clearly showing an involvement of SerpinA3K in renal injury. Moreover, the authors demonstrate that the attenuation of renal damage in KOSA3 mice is mediated by antioxidant pathways, as confirmed by improved levels of Sod and Cat enzymes. The work is novel, very interesting and well presented. The English language is very good. However, I have many major concerns listed below that should be carefully addressed by the authors before I consider their work suitable for publication.

Major

·      Maybe material and methods sections could be added before results 

·      Pag 10 line 313. Were mice used for experiments male or female? This should be specified in materials and methods.

·      The anesthetic sodium pentobarbital was given mice at lower concentration for GFR measurement (30 mg/kg) than the one used for renal ischemia induction (50 mg/kg). Maybe this was needed to avoid a great decrease in blood pressure in animals during GFR measurement? Please, clarify this.

·      Pag 12 line 386. The concentration of HRP b actin antibody for western blot is indicated as 1:1000,000. It seems to me to higher concentration for an antibody, was it a mistake? Please, clarify it.

·      Figure 1a. I would suggest to add the base pair (BP) numbers close to each band in the electrophoretic gel instead to put in the table. Moreover, if possible, the image should also show the specific used ladder. 

·      Figure 1d. It seems that the survival rate curve is shown only for WT mice but not for KO mice. Indeed, only a black line is shown in the graph. Please, clarify this. 

·      Figure 2. Please, specify in figure legend the meaning of S (i.e. WT + S, KOSA3 + S etc) as well as KW and BW.

·      Figure 2 F. The GFR measurements are shown only for WT + IR and KOSA3 +IR, which are certainly the most important groups. Maybe the manuscript would be core complete if showing the GFR values in the other 2 studied groups (wt + S and KOSA3 +S) or at least in the WT +S. Do the authors have these data? Otherwise comment this point. 

·      I suggest the authors, when discussing about the tubular damage and renal injury induced by IR, to cite this recent article (DOI: 10.1007/s00424-022-02686-8) where the authors characterize tubular damage and SNGFR in both rats and mice. In this way, the current study will result more updated and complete. 

·      Figure 4 b. The signal of SerpinA3k is not very clear neither in WT+S nor in WT+IR groups. It is also difficult to recognize the renal tubules that are barely visible. Maybe the images should be acquired again with better setting (or more focused) or just modified in terms of contrast and /or sharpness to better show the tubules as well as the cytosolic (WT+S) and apical (WT+IR) signals.

Minor

·      Pga12 line 400. “Was used to perform” instead of “was used to performed” 

·      Pag 3 line 105. I think the authors refer to Figure 2D and 2E (not 2F). please, correct the mistake.

·      Figure 3 A, B, C, D. I suggest to write the respective groups above the images for a quicker interpretation of the data. Moreover, please specify the meaning of “ATI” in figure legend.

·      Figure 4. Please, specify in figure legend how many hours post IR the sections were labeled for immunofluorescence.

Author Response

Reviewer #1:

We are very grateful for the comments and suggestions made by the reviewer that have improved the new version of our manuscript. 

Major Concerns

  • Maybe material and methods sections could be added before results 

            Thank you for your suggestion, however, the sections order is established for the journal.

  • Pag 10 line 313. Were mice used for experiments male or female? This should be specified in materials and methods.

            Sorry for omitting this important information. The study was performed in male mice; this information was added in the last paragraph of page 10.

  • The anesthetic sodium pentobarbital was given mice at lower concentration for GFR measurement (30 mg/kg) than the one used for renal ischemia induction (50 mg/kg). Maybe this was needed to avoid a great decrease in blood pressure in animals during GFR measurement? Please, clarify this.

            We appreciate your observation that allows us to clarify this issue as follows: This dose was chosen to keep mice sedated but without impact on GFR. This sentence was added in 5.2 section on page 11.

  • Pag 12 line 386. The concentration of HRP b actin antibody for western blot is indicated as 1:1000,000. It seems to me to higher concentration for an antibody, was it a mistake? Please, clarify it.

            HRP b-actin antibody manufactured by Abcam is extremely efficient to detect b-actin in the kidney, together with the high sensitivity captured by the iBright equipment. Therefore, we have used this dilution in this and other studies already published:

Sirtuin 7 Deficiency Reduces Inflammation and Tubular Damage Induced by an Episode of Acute Kidney Injury-- Andrea Sánchez-Navarro, Miguel Ángel Martínez-Rojas, Adrián Albarrán-Godinez, Rosalba Pérez-Villalva, Johan Auwerx, Abigail de la Cruz, Lilia G. Noriega, Florencia Rosetti, Norma A. Bobadilla

Vegfa promoter gene hypermethylation at HIF1α binding site is an early contributor to CKD progression after renal ischemia. Andrea Sánchez-Navarro, Rosalba Pérez-Villalva, Adrián Rafael Murillo-de-Ozores, Miguel Ángel Martínez-Rojas, Jesús Rafael Rodríguez‐Aguilera, Norma González, María Castañeda-Bueno, Gerardo Gamba, Félix Recillas-Targa, Norma A. Bobadilla Sci Rep. 2021; 11: 8769. Published online 2021 Apr 22. doi: 10.1038/s41598-021-88000-5.

Repeated Episodes of Ischemia/Reperfusion Induce Heme-Oxygenase-1 (HO-1) and Anti-Inflammatory Responses and Protects against Chronic Kidney DiseaseJuan Antonio Ortega-Trejo, Rosalba Pérez-Villalva, Andrea Sánchez-Navarro, Brenda Marquina, Bernardo Rodríguez-Iturbe, Norma A. Bobadilla Int J Mol Sci. 2022 Dec; 23(23): 14573. Published online 2022 Nov 23. Int J Mol Sci. 2022 Mar; 23(5): 2573. Published online 2022 Feb 25.

  • Figure 1a. I would suggest to add the base pair (BP) numbers close to each band in the electrophoretic gel instead to put in the table. Moreover, if possible, the image should also show the specific used ladder.

The ladder used is now included in Figure 1a and the base pair (BP) numbers are close to each band.

  • Figure 1d. It seems that the survival rate curve is shown only for WT mice but not for KO mice. Indeed, only a black line is shown in the graph. Please, clarify this. 

The survival rate is the same for WT and KO mice, so they are overlapped. To fix this, the lines have been slightly separated and thickened in this graph.

  • Figure 2. Please, specify in figure legend the meaning of S (i.e. WT + S, KOSA3 + S etc) as well as KW and BW.

Thanks for your observation. The meanings were added in the Figure legend. 

  • Figure 2 F. The GFR measurements are shown only for WT + IR and KOSA3 +IR, which are certainly the most important groups. Maybe the manuscript would be core complete if showing the GFR values in the other 2 studied groups (wt + S and KOSA3 +S) or at least in the WT +S. Do the authors have these data? Otherwise comment this point. 

            The Figure 2F considers the GFR for the four studied groups. Since to make more evident visually the different response between the WT and KO mice induced by the ischemic injury, the percentage change experienced by the WT+IR with respect to the value of the WT group was plotted, the same for KOSA+IR mice with respect to the KOSA group without IR.

  • I suggest the authors, when discussing about the tubular damage and renal injury induced by IR, to cite this recent article (DOI: 10.1007/s00424-022-02686-8) where the authors characterize tubular damage and SNGFR in both rats and mice. In this way, the current study will result more updated and complete.

We appreciate that you share us this novel technique, we have added the following paragraph in the discussion section: It would even be interesting to evaluate in future studies, the tubuloglomerular feedback, the SNGFR and the morphology of the proximal tubular epithelium in these groups of animals, especially since the SNGFR and the morphology can now be evaluated by a new technique such as linescan multiphoton microscopy (Costanzo, et. al,).

  • Figure 4 b. The signal of SerpinA3k is not very clear neither in WT+S nor in WT+IR groups. It is also difficult to recognize the renal tubules that are barely visible. Maybe the images should be acquired again with better setting (or more focused) or just modified in terms of contrast and /or sharpness to better show the tubules as well as the cytosolic (WT+S) and apical (WT+IR) signals.

An apology if the photomicrograph is not clear in the pdf that was generated by the journal platform, we are attaching the microphotograph alone, where the differences in the immunofluorescence among the groups can be clearly seen. It is very likely that these microphotographs will look correctly in the final publication.

Minor

  • Pga12 line 400. “Was used to perform” instead of “was used to performed”. Thanks, it was corrected in the first paragraph of page 13.
  • Pag 3 line 105. I think the authors refer to Figure 2D and 2E (not 2F). please, correct the mistake. It was corrected
  • Figure 3 A, B, C, D. I suggest to write the respective groups above the images for a quicker interpretation of the data. Moreover, please specify the meaning of “ATI” in figure legend. Thank you for these observations, all of them were corrected
  • Figure 4. Please, specify in figure legend how many hours post IR the sections were labeled for immunofluorescence. This issue was corrected in the legend.

Reviewer 2 Report

The work by González-Soria et al on SerpinA3K Deficiency Reduces Oxidative Stress in Acute Kidney Injury is an extremely interesting literature item and requires only a few editorial changes. First of all, I would like to ask you to use citations in the text in accordance with the journal format, i.e. [3-6] and not [3], [4], [5], [6], and to keep a space before placing the citation. Bradoz, please increase the resolution and size of figure 4 (especially the photos of gels and give the scale to 4B), the same applies to figures 5 and 6. in the current version they are difficult to read for the reader. There is also a lack of extremely valuable information regarding: Author Contributions, Institutional Review Board Statement, Data Availability Statement, etc.

The Conclusion part is also missing, please add it, it will allow you to summarize the most important achievements of this work

Author Response

Reviewer #2:

We are very grateful for the comments and suggestions made by the reviewer that have improved the new version of our manuscript.

The work by González-Soria et al on SerpinA3K Deficiency Reduces Oxidative Stress in Acute Kidney Injury is an extremely interesting literature item and requires only a few editorial changes. First of all, I would like to ask you to use citations in the text in accordance with the journal format, i.e. [3-6] and not [3], [4], [5], [6], and to keep a space before placing the citation. Bradoz, please increase the resolution and size of figure 4 (especially the photos of gels and give the scale to 4B), the same applies to figures 5 and 6. in the current version they are difficult to read for the reader. There is also a lack of extremely valuable information regarding: Author Contributions, Institutional Review Board Statement, Data Availability Statement, etc.

The Conclusion part is also missing, please add it, it will allow you to summarize the most important achievements of this work

Thank you for your comments. We have corrected the citation format. The resolution of each image was decreased when the PDF was generated by the platform of the journal, therefore, we are attaching the immunofluorescence images in pdf and the images of each blot are in the supplementary material, in both the images have good resolution. The missing editorial fields have been corrected.

Reviewer 3 Report

This is an interesting study depicting the role of SerpinA3K in renal oxidant response in Acute Kidney Injury. 

There are some minor comments to further improve the manuscript. 

1. Please look carefully through the manuscript and write H2Olike this throughout. Sometimes it is written like H2O2. 

2. in line 25, it should be HIF1a

3. in line 33, it should be evident instead of evidenced

4. in figure 1C, why do you see SerpinA3K bank in the last lane? Do you have any explanation for that?

5. Please add statistics in all the bar charts (some of the figures are missing it). Even if it is non-significant, it is good to have it in the figure. 

6. in line 177, it should be UH2O2

7. How would be the changes in the protein levels of Catalase, SOD2 and GPX1 in different mouse models? The regulation can be at transcriptional or translational levels. You have evaluated the mRNA levels in Fig 6, but it would be also an important addition to evaluate the protein levels as well. 

8. the original blots are sometimes missing any labelling of protein ladder. So, it should be added to see the size of a protein. 

Author Response

Reviewer #3:

We are very grateful for the comments and suggestions made by the reviewer that have improved the new version of our manuscript.

This is an interesting study depicting the role of SerpinA3K in renal oxidant response in Acute Kidney Injury. 

There are some minor comments to further improve the manuscript. 

  1. Please look carefully through the manuscript and write H2Olike this throughout. Sometimes it is written like H2O2. 

            Sorry for this error, it has already corrected throughout the manuscript.

  1. in line 25, it should be HIF1a

            Thank you for your observation. Now, it has been corrected.

  1. in line 33, it should be evident instead of evidenced

Agree with this issue and it was corrected.

  1. in figure 1C, why do you see SerpinA3K bank in the last lane? Do you have any explanation for that?

We appreciate your feedback, although a band is noticeable of smaller size in one of the KO mice, we consider that it could be a non-specific band or contamination, but that it does not even resemble the real expression of serpinA3K in the WT and HT mice.

  1. Please add statistics in all the bar charts (some of the figures are missing it). Even if it is non-significant, it is good to have it in the figure. 

Thank you for your comment. We added in all graphs only when the differences were significant, we believe that the information of not significant differences, distract the reader from the relevant findings.

  1. in line 177, it should be UH2O2

This mistake has been corrected.

  1. How would be the changes in the protein levels of Catalase, SOD2 and GPX1 in different mouse models? The regulation can be at transcriptional or translational levels. You have evaluated the mRNA levels in Fig 6, but it would be also an important addition to evaluate the protein levels as well. 

            Thanks for your comment. Our hypothesis is that the expression of antioxidant enzymes can be regulated at transcriptional and translational level. However, we consider that given the short term in which the study was carried out (24 hours after the IR) it is unlikely that we will see significant changes in protein levels. In future studies, where the temporal course of the expression of antioxidant enzymes will be evaluated, it is very likely that the increase in protein levels can be verified.

  1. the original blots are sometimes missing any labelling of protein ladder. So, it should be added to see the size of a protein. 

We have identified 3 original blots with no protein ladder present. Specifically, the beta-actin blot for the PGC-1alpha analysis. In this particular case, due to the high interference caused by our internal markers in the capture equipment, it was necessary to trim the ladder band. Nevertheless, the two sections of each blot are shown to verify the match between the target protein and the control protein, and we add the previous images in order to increase the level of confidence.

Round 2

Reviewer 1 Report

The authors have carefully addressed all my concerns, thank you. I strongly suggest this article for publication. Well done!!!

Please, only correct the position of letters E,F G,H in figure 3; letters D,E in figure 4. Moreover, remove letter H from figure legend f figure 4.